# Folate Receptor Alpha—A Novel Approach to Cancer Therapy

**DOI:** 10.3390/ijms25021046

**Published:** 2024-01-15

**Authors:** Teresita Gonzalez, Meri Muminovic, Olger Nano, Michel Vulfovich

**Affiliations:** Memorial Cancer Institute, Pembroke Pines, FL 33028, USA; mmuminovic@mhs.net (M.M.); onano@mhs.net (O.N.); mvulfovich@mhs.net (M.V.)

**Keywords:** folate receptor alpha, antibody drug conjugate, FOLR1

## Abstract

Folate receptor α (FR) was discovered many decades ago, along with drugs that target intracellular folate metabolism, such as pemetrexed and methotrexate. Folate is taken up by the cell via this receptor, which also targeted by many cancer agents due to the over-expression of the receptor by cancer cells. FR is a membrane-bound glycosyl-phosphatidylinositol (GPI) anchor glycoprotein encoded by the folate receptor 1 (FOLR1) gene. FR plays a significant role in DNA synthesis, cell proliferation, DNA repair, and intracellular signaling, all of which are essential for tumorigenesis. FR is more prevalent in cancer cells compared to normal tissues, which makes it an excellent target for oncologic therapeutics. FRα is found in many cancer types, including ovarian cancer, non-small-cell lung cancer (NSCLC), and colon cancer. FR is widely used in antibody drug conjugates, small-molecule-drug conjugates, and chimeric antigen-receptor T cells. Current oncolytic therapeutics include mirvetuximab soravtansine, and ongoing clinical trials are underway to investigate chimeric antigen receptor T cells (CAR-T cells) and vaccines. Additionally, FRα has been used in a myriad of other applications, including as a tool in the identification of tumor types, and as a prognostic marker, as a surrogate of chemotherapy resistance. As such, FRα identification has become an essential part of precision medicine.

## 1. Introduction

### 1.1. Importance of Folate Metabolism

Throughout the world, the incidence and mortality of cancer have been growing rapidly, making it one of the leading causes of death. Cancer mortality has contributed to the stagnation of increases in life expectancy in every country in the world. [1] The advent of genomics, proteomics and new scientific methods, which are now aided by artificial intelligence (AI), has led to the development of new approaches to fighting cancer. Beyond chemotherapy, immunotherapy and antibody drug conjugates are now important treatment approaches for most malignancies. Finding new targets and homing in on new clinical developments are critical tasks that require fresh insights into new or known targets.

Folate, also known as vitamin B9, plays a significant role in human development, as well as in tumorigenesis. It is essential for transporting one-carbon units, for the DNA and RNA synthesis needed for cell division, and for other processes such as neural-tube development [2]. The active form of folate is tetrahydrofolic acid, while the form that circulates in the blood is 5-methyltetrahydrofolate. B9 is an essential nutrient that, if deficient, can lead to numerous conditions such as anemia and neural-tube defects in pregnancy. The folate receptor has been under investigation in oncology as a potential target for new therapeutic agents. Many cancers have abnormal receptor-expression patterns, while normal tissues generally do not exhibit these patterns. This difference makes it possible to target cancer cells without affecting healthy cells. Hence, targeting this receptor is of utmost significance, as it could allow for more targeted killing of cancer cells, optimizing treatment effectiveness and minimizing off-target effects.

Exploration of this receptor is important because it may represent a new and unique target for treating cancer. This review article provides an overview of the receptor, its importance, its role in various cancer types, current therapies, and novel ways it is being researched for use in other treatment approaches, with the hope of guiding treatment in the future.

### 1.2. FRα Tissue Distribution, Function and Significance in Tumorigenesis

The importance of the folate receptor (FR) was first discovered when the elimination of FOLR1 caused the death of mice, indicating its vital role in development [3]. FR exists in four isoforms—FRα, FRβ, FRδ and FRγ—which are distributed across different tissues. Among these, FRα has been studied the most [4,5]. FRα is a membrane-bound, glycosyl-phosphatidylinositol (GPI) anchor glycoprotein that is encoded by the FOLR1 gene found in chromosome 11 [6]. FRα has a high affinity for folic acid, as well as for other reduced folates such as 5-methyltetrahydrofolate and tetrahydrofolate. FR binds to oxidized folate more strongly and transports it to the cytoplasm via potocytosis. The acidic pH dissociates the folate from the receptor and releases it into the cytoplasm [7].

FR plays a significant role in various processes that are essential for tumorigenesis, such as DNA synthesis, cell proliferation, DNA repair, and intracellular signaling [8]. For instance, folate binding to FRα leads to a series of intracellular signaling cascades via phosphorylation, which activates the ERK, STAT3 pathway, which leads to important the activation of regulatory mechanisms involved in cell growth, as illustrated in Figure 1. Additionally, FRα assists tumor invasion and spread by down-regulating adhesion molecules such as E-cadherin [9].

FR is a protein that is found in various tissues of the body, such as the kidneys, lungs, intestines, and central nervous system [10]. However, the receptor does not have easy access to folate in normal tissues because it is located in the apical portion of cells, making it less accessible to circulating folate. In contrast, tumor cells have direct access to folate, and FRα (the alpha subtype of FR) is highly expressed in ovarian cancer cells, for example [11]. This high expression of FRα in cancer cells compared to normal tissues makes it an ideal target for cancer treatments. For instance, FR is expressed in 14–74% of non-small cell lung cancers, 72–100% of mesotheliomas, 20–50% of endometrial cancers, 35–68% of triple-negative breast cancers, and 76–89% of ovarian cancers. FR expression in non-malignant tissues is limited. For example, FR expression is 0–25% in normal ovarian tissue and 0–20% in non-malignant breast tissue [12]. The specificity of FR expression in cancer cells allows for the use of targeted therapies, such as antibody-drug conjugates (ADC), CAR-T cell therapy, and vaccines, among other agents, to target cancer cells without affecting most healthy cells. Table 1 and Figure 2 provide a summary of the different expression patterns of FR across various cancer types.

## 2. Role of FRα in Ovarian Cancer

FRα plays a significant role in ovarian cancer (OC) due to its high expression in ovarian tumor cells compared to its limited expression in non-malignant ovarian cells. Compared to other cancer types, ovarian cancer cells show higher expression of FRα. Hence, there is a growing interest in targeting this receptor in OC, especially with the approval in 2022 of the first ADC utilizing FRα, mirvetuximab soravtansine (MIRV), which is approved for use in FRα-positive, platinum-resistant epithelial ovarian, fallopian-tube, or primary peritoneal cancer.

FRα expression has a linear correlation with stage and grade in OC, with different expression patterns across subtypes [13]. In a study analyzing FR expression in OC, FRα expression was apparent in 72% of primary and 81.5% of recurrent OC. In primary OC, expression was higher in the serous OC subtype (81.7% vs. 59.8% in other tumor types; 66.7% in the endometrioid subtype and 63.3% in the clear-cell subtype). Moreover, high-grade tumors were more likely to express FRα than were low-grade tumors (75.8% vs. 48%) [14]. The role of FRα in tumorigenesis in OC is partially explained by the down-regulation of tumor suppressor caveolin-1 (cav-1), as various studies show an inverse relationship between cav-1 and FRα levels. OC cells express high levels of FRα and low levels of cav-1 [15]. Cav-1 is involved in signal-transduction pathways that are important in regulating the proliferation of OC cells [16].

Targeting the FRα receptor in OC is a novel tool for treating gynecologic malignancies. It is important because it can be used to avoid unwanted side effects and for more effective tumor killing. Understanding the preferential expression of FRα in OC compared to normal cells would be useful in developing better treatment algorithms.

Changes in FRα levels have been observed in patients with ovarian cancer who have undergone treatment. Measurement of these changes could yield a useful method of detecting residual disease in the future. In a study conducted by Bax et al., the expression of FRα was evaluated before, during, and after neo-adjuvant chemotherapy using immunohistochemistry [17]. This study included 316 patients with ovarian cancer, and it was found that FRα was present in about half of them. Higher levels of FRα were observed before any systemic treatment, and lower levels were seen in treated tumors. This result underscores the importance of monitoring patient responses to treatment. In cases in which minimal residual disease is present, clinicians can predict poor treatment outcomes and make appropriate changes in therapy.

This powerful receptor-guided treatment can be useful not only in ovarian cancer, but also potentially in increasing the effectiveness of drugs using antibody-drug conjugates (ADCs). Recent studies have investigated the use of this important marker in the field of gynecologic malignancies, particularly in relation to ADCs. One such novel ADC targeting FRα is STRO-002. In a study conducted by Li et al., STRO-002 was found to bind to FRα with increased affinity, leading to the avid internalization of the ADC into cells carrying FRα [18]. After this process occurs, a cytotoxin is released, specifically a tubulin-targeting cytotoxin called 3-aminophenol hemiasterlin, which was seen to significantly reduce tumor growth, particularly when it was combined with platinum- or Avastin-based treatments. These findings have paved the way for current phase 1 studies such as NCT03748186 and NCT05200364.

The concept of using ADC has gained popularity in the field. Recently, several early phase 1 studies have contributed to the idea of using targeted agents and improved drug-delivery systems. Elu001, for example, represents a groundbreaking development. It incorporates about 22 molecules of the topoisomerase-1 inhibitor exatecan, with about 13 folic-acid-targeting moieties. This unique agent achieves better penetration into tumor cells compared to ADCs [19]. Moreover, it ensures rapid renal elimination, which is crucial in reducing toxic side effects. Once it binds to the receptor on the outside of the tumor cells, it internalizes into the tumor and goes to the lysosome, which releases the exatecan. This drug-delivery system is not only interesting, but also has many applications beyond gynecologic malignancies. An ongoing study is exploring other malignancies that may express the receptor, such as cholangiocarcinoma lung, gastric, triple-negative breast, and colorectal malignancies, among others [19].

There have been recent studies that highlight the importance of FRα not only in gynecologic malignancies, but also in other gynecologic pathologies. One such study conducted by Wang et al. found that higher expression was seen in endometriosis tissue samples from the fallopian tubes and ovaries as compared to paired normal endometrial samples [20]. In normal ovarian tissue, negative expression patterns were seen. This finding suggests the potential of developing alternative and novel methods to identify pathology within the gynecologic system that may not be limited to the field of oncology.

A study conducted by Saito et al. examined resected gynecologic carcinosarcomas and analyzed the expression of FRα and its relationship with HER2 [21]. This study found that all 120 patients with uterine carcinosarcoma expressed FRα, and over 30% of patients who had high FRα expression were HER2-negative. Additionally, all patients with ovarian carcinosarcoma also expressed FRα. However, this study did not find any significant relationship between FRα expression and prognosis. Nevertheless, this study highlights the importance of this receptor in other gynecologic subtypes like carcinosarcoma.

The idea of using more targeted agents in rare gynecologic malignancies has generated a great deal of interest in the medical field. One example of this approach is the use of eribulin ADCs, which have been previously used to minimize toxicity while maximizing treatment efficacy in rare gynecologic cancers such as ovarian carcinosarcoma, uterine serous carcinoma, and ovarian clear-cell adenocarcinoma [22]. Farletuzumab ecteribulin is an ADC that uses using this method, with MORAb-109 targeting eribulin to mesothelin. This approach has been studied in newer trials, as shown by Vandenberg et al. [22]. This idea is important because many of these rare gynecologic malignancies have poor overall responses, and finding novel ways to treat them can offer hope for our patients.

In this article, we will address the advent of MIRV based on the SOYAYA study, which has made significant clinical progress in bypassing platinum-resistant ovarian cancer. There is an unmet need for effective treatment of patients with platinum-resistant ovarian cancer, and many studies are being conducted to expand on this idea. For instance, Luveltamab tazevibulin (luvelta) is a novel FRα-targeting ADC with a hemiasterlin warhead (DAR4) that showed efficacy in a phase 1 study in 32 patients with relapsed or advanced epithelial ovarian cancer with broad FRα expression, with an ORR of 37.5% [23].

In a phase 1/2 study of rinatibet sesutecan (Rina-S) known as the PRO1184-001 study, another new ADC targeting FRα was tested in patients with locally advanced and/or metastatic solid tumors like ovarian tumors, NSCLC, breast and endometrial tumors, or mesothelioma. This first-in-human trial (NCT05579366) showed that the novel ADC has a relatively safe profile and promising efficacy [24].

The use of FRα expression and targeting through ADC has established a promising future direction for oncology, particularly in the field of gynecologic malignancies. It offers the potential to enhance safety and efficacy, even in platinum-resistant, advanced, or rare malignancies. This approach could pave the way for treating other malignancies that express FRα, which would be a revolutionary development in cancer treatment.

## 3. Role of FRα in Other Cancer Types

It has been observed that many types of cancer exhibit over-expression of FRα. A study of 320 surgical NSCLC tissues revealed that adenocarcinomas, especially epidermal growth factor receptor (EGFR) mutants, had higher expression of FRα compared to squamous-cell carcinomas. Tumors in non-smokers and early-stage tumors also had higher expression of FRα, suggesting that it may be a promising target in NSCLC [25]. In another study of colon cancer, FRα positivity was higher in carcinomas (44% in metastatic and to 33% in primaries) compared to normal colon tissue or adenomas (7%) (*p* < 0.001) [26]. Other cancers with high FRα expression include mesotheliomas, gynecologic malignancies, breast cancers, and head and neck cancers. These findings are important, as they expand the scope of potential treatment applications from gynecologic malignancies to many other cancer subtypes.

Numerous studies have explored the use of the FRα receptor across various types of cancer. Cao et al. conducted research to determine whether FRα could be used in a liquid-biopsy diagnostic technique for gastric cancer (GC). They found that FRα expression correlated with poor prognosis in lympho-vascular invasion or extra-nodal disease and served as a useful tool for screening patients in a non-invasive way, outperforming the CEA antigen with sensitivities above 80% [27]. Miner et al. investigated the use of FRα-targeted imaging techniques in gliomas, proposing new ways of imaging and targeting gliomas in the future [28]. This study found that FRα was upregulated in gliomas in mouse models and human glioblastoma tissues. The PRO1184 study investigated the receptor’s role in various tumor types such as ovarian, NSCLC, breast, and endometrial tumors, as well as in mesotheliomas (NCT05579366). Kumari et al. researched the role FRα plays in triple-negative breast cancer (TNBC), a highly aggressive subtype of breast cancer with high rates of metastasis, poor prognosis, and relapses [29]. Similarly, Necela et al. investigated its role in TNBC. They analyzed the distribution of the receptor across various types of breast cancer, including hormone-receptor-positive cancer, HER2+ cancer, and TNBC, and found that expression varied across all subtypes studied. The TNBC tissues had the highest expression of FRα, while hormone-receptor-positive breast tumors had the lowest [30]. Interestingly, FRα expression did not correlate with tumor stage or nodal status. However, loss of the receptor in knockout models inhibited tumor growth. Thus, agents blocking this receptor may potentially be therapeutically useful for TNBC patients.

Several studies have utilized FRα as a tool to deliver nanoparticles and overcome treatment resistance. For instance, Singh et al. developed novel nanoparticles encapsulated with resveratrol (RES) and docetaxel and conjugated with folic acid on the surface. These nanoparticles lowered resistance to docetaxel in prostate-cancer tissues by inhibiting drug-efflux pathways [31]. This approach could potentially help treat patients with prostate cancer who have relapsed on taxol-based therapy.

Bhattacharya et al. recently demonstrated that targeting the folate receptor in oral cancers with chemotherapeutic drugs could be promising [32]. Clinical samples of oral squamous-cell carcinoma showed higher expression of FRα. Glucose-derived carbon nanospheres primed with folate-based cationic lipid and doxorubicin were utilized on oral squamous cell cancer cells in vitro, and showed both good killing activity and better drug delivery because the drug product was retained for longer periods inside of the tumor cell compared to drugs delivered without the nanospheres.

The delivery of nanoparticles using FRα has shown similar results in other subtypes of solid tumors. Prajapat et al. found that delivery of immunotherapeutic and targeted agents using nanoparticle technology has promise in reducing the loss of appropriate targeting and specificity for tumor cells in melanomas [33]. Wang et al. showed that not only is FRα useful for diagnosing lung cancer, but it can also be useful in detecting lung-cancer recurrence [34]. FRα levels correlated to worse progression-free survival outcomes.

Chen et al. conducted a systematic review analyzing 11 studies with 3469 patients to investigate the diagnostic value of circulating tumor cells (CTCs) expressing FRα in lung cancer. This study showed a pooled sensitivity and specificity of 0.79 (95% CI 0.76, 0.82) and 0.84 (95% CI 0.81, 0.96), respectively [35]. This result suggests that levels of CTCs expressing FRα can be an excellent tool for detecting lung cancer.

## 4. Current Therapies Targeting FRα

### Mirvetuximab Soravtansine (IMGN853)

Chemotherapy drugs such as methotrexate and pemetrexed have traditionally used the folate pathway to treat cancer. However, recent studies have shown that targeting FRα is a more effective approach. Despite this target having been discovered in the 1990s, only one FRα-targeted treatment has been approved to date. Many clinical trials, such as those of farletuzumab in 2013 and Vintafolide in 2014, failed to meet their primary endpoints. It was not until 2022 that significant progress was made in creating the first ADC using FRα. On 14 November 2022, the FDA granted accelerated approval to Mirvetuximab soravtansine (MIRV) for the treatment of patients with FRα-positive, platinum-resistant epithelial ovarian, fallopian-tube, or primary peritoneal cancer who had already received one to three systemic treatment regimens. MIRV, also known commercially as ELAHERE, is administered intravenously every three weeks at a dose of 6 mg/kg, adjusted for ideal body weight, until disease progression or unacceptable toxicity occurs. This approval was based on the results of the SOYAYA study, which examined high-FRα-expressing, platinum-resistant epithelial ovarian, fallopian, or primary peritoneal cancers in patients who had received one to three previous systemic therapies [36]. The assay used in this study was the Ventana FOLR1 assay. MIRV is an antibody-drug conjugate (ADC) that consists of an FRα-binding antibody and a maytansinoid payload called DM4, which is a tubulin inhibitor. MIRV was designed to target and kill FRα-positive cancer cells, as illustrated in Figure 3. MIRV was the first antibody-drug conjugate approved for the treatment of platinum-resistant ovarian cancer.

The SOYAYA study was a large-scale clinical study that involved 106 patients with FRα-positive tumors, including high-grade serous epithelial OC, fallopian, or primary peritoneal cancer. The Ventana FOLR-2.1 assay was used to diagnose FRα-positive tumors by using a qualitative immunohistochemical assay, which detects moderate-to-strong membrane staining in at least 75% of tumor cells. Patients who participated in this study had an ECOG performance status of 0 to 1, had undergone major surgery at least four weeks before the first dose of MIRV, and had received one to three prior lines of anticancer therapy, including one therapy containing bevacizumab. Patients were treated with 6mg/kg of MIRV administered intravenously every three weeks. This study found that five patients achieved complete response (CR), with a median response duration of 6.9 months (95% CI: 5.6–9.7) and an overall response rate (ORR) of 31.7% (95% CI: 22.9–41.6%) [36]. The most common side effects were blurry vision, nausea, and keratopathy. The more serious side effects of MIRV included ocular toxicity, pneumonitis, peripheral neuropathy, and fetal toxicity. However, most of the side effects were tolerable gastrointestinal symptoms and electrolyte imbalances. This study demonstrated a breakthrough targeted treatment for OC using FRα and set a new standard of care.

The MIRASOL Phase III study, presented at the American Society of Clinical Oncology (ASCO) 2023 conference, validated the findings of the SOYAYA study. This study involved 453 OC patients with high FRα expression and one to three prior treatments who were randomized to receive MIRV or IC chemotherapy with paclitaxel, pegylated liposomal doxorubicin, or topotecan. The median follow-up period was 13.1 months. In the bevacizumab-pretreated subset, PFS HR was 0.64 (95% CI: 0.492–0.842) and OS HR was 0.74 (95% CI: 0.535–1.036); in the bevacizumab-naïve subset, PFS HR was 0.66 (95%Chi CI: 0.459–0.942) and OS HR was 0.51 (95% CI: 0.306–0.860). PFS with MIRV was 5.62 months, compared to 3.98 months in the IC arm. In this study, 12 patients had a complete response (CR), and the ORR was 42.3% [37]. The side effects of MIRV were consistent with those found in the SOYAYA study, with mild gastrointestinal side effects and lower rates of grade 3 or severe adverse effects than were seen in the IC arm. This study confirmed the safety and efficacy of MIRV and established a new standard of care for OC treatment. Future studies such as PICCOLO (IMGN853-0419 and GLORIOSA (NCT05445778) may expand the indications for MIRV.

Table 2 below summarizes recent clinical trials utilizing FRα. Some of these trials are further investigating and expanding on applications of MIRV, as seen below, while others are investigating its use in vaccines and imaging techniques.

## 5. Utilization of FRα in CAR-T Therapy

Chimeric antigen receptor (CAR)-T cell therapy is a new frontier in cancer treatment that shows great promise for the future. To ensure better outcomes for patients with most cancers, it is important to optimize CAR-T therapy by reducing its adverse effects and creating an effective delivery system. Ongoing clinical trials have studied the use of FRα in CAR-T therapy, as FRα is found in many cancer types and has limited expression in normal tissues, making it ideal for targeted killing.

The use of CAR-T and T-cell engagers (TCEs) has been rapidly increasing. However, one of the challenges in implementing T-cell therapy widely is the off-target effects caused by antigen expression in normal tissues, which can lead to unexpected and unwanted inflammatory responses. Optimizing this form of treatment by minimizing off-target effects is an important challenge. Several studies have explored the potential of leveraging our immune system to selectively destroy cancer cells while sparing us unwanted side effects. If FRα is selectively expressed at higher levels in malignant tissues, then it would make sense to capitalize on this difference in CAR-T therapy. Avanzino et al. investigated this idea using TNB-928B, a fully humanized TCE with a bivalent arm for FRα, to target FRα-over-expressing cancer cells, as seen in Figure 4 [38]. This agent causes selective targeting of high-expressing tumor cells with the added benefit of decreasing cytokine release. This agent demonstrated ex vivo cancer-killing effects in mouse models of ovarian cancer.

Despite the success of CAR-T therapy, relapses are inevitable, and resistance patterns have been observed. Resistance can occur via various pathways, such as the tumor microenvironment causing immunosuppression, mutations in the tumor cells conferring resistance, defects in the CAR-T product, or the patient’s immune system being defective and not highly effective at tumor killing [39]. Proposed mechanisms to overcome this problem include fixing the product itself, creating targets for new/other antigens, and targeting the tumor microenvironment.

Some studies have investigated resistance patterns of FRα-CAR T cells by analyzing markers such as levels of carcinoma-associated fibroblasts (CAFs), which regulate cancer progression through IL-6, which in turn can promote PD-L1 expression, which is linked to FRα-CAR T resistance/ineffective CAR T killing in patients with breast cancer receiving doxorubicin [40]. These complex studies are expanding our knowledge, allowing us to find ways to overcome treatment resistance.

In 2006, a phase I study was the first to use adaptive immunotherapy with gene-modified autologous T cells and FR to treat metastatic ovarian cancer [41]. In this study, five patients experienced grade 3 and grade 4 treatment-related toxicities, probably due to IL-2 administration. T cells were detected in the circulation for 2 days but declined after 1 month to undetectable levels because an inhibitory factor developed, reducing the ability of the T cells to effectively target FR-positive cells. This finding highlights the need to improve on outcomes of CAR-T therapy.

According to an abstract published in ASCO 2022, a study evaluating IL-2 independent effector function by anti-FOLR1 CoStAR T cells (ITIL-306) in vitro found that tumor growth was significantly reduced and survival was improved by this treatment, with five out of six mice alive at day 96 [42]. Although there are many obstacles to the use of this receptor in treatments and imperfections in its application, it has been shown in many studies that the use of this receptor can pave the way to success. The success of these in vitro studies has prompted the investigation of the effectiveness of using anti-FOLR1 T cells to treat other cancer types. For example, ITIL-306-201 is an ongoing phase 1a/1b multicenter clinical trial investigating the safety and feasibility of using ITIL-306 in adults with advanced solid malignancies that have progressed after standard therapy [43]. Ultimately, more research is needed in this field to improve the efficacy of immunotherapy and CAR-T.

## 6. Promising Vaccine Studies

Given the successful use of CAR-T targeting FRα, it is not surprising that FRα can be utilized in other immunotherapy approaches. Vaccines can potentially induce our immune systems to produce antibodies against FRα. If FRα is expressed mainly in cancer cells, rather than in healthy tissues, and higher expression levels correlate to more severe disease and more aggressive cancers, it would make sense to train our immune system to block this receptor after remission is achieved. This approach can theoretically be used as a maintenance therapy to prevent disease relapse by depriving tumors of their essential food source. This concept has been explored in cancer vaccines, and early studies have already confirmed the safety of this approach. Future research will focus on its efficacy.

The safety of vaccination against FRα has been previously studied. In a phase 1 clinical trial that tested safety and immunogenicity in patients with ovarian cancer and breast cancer who had achieved remission, vaccination utilizing a multi-epitope FRα peptide vaccine was considered safe, producing mostly injection-site reactions and showing no association with grade 3 toxicity [44]. Establishing safety is always the first step in testing for efficacy. The effectiveness of this approach has been sparking interest. Vaccinations could also potentially be efficacious in augmenting the effects of other immunotherapies. For example, studies have utilized vaccination in combination with checkpoint inhibitors to increase T cell responses in patients. A study examining the use of a FRα peptide vaccine in combination with the programmed death ligand 1 (PD-L1) inhibitor durvalumab in patients with platinum-resistant ovarian cancer showed increased T cell responses in all 27 patients [45]. This finding is revolutionary because it shows that the treatment can potentiate the effects of existing therapies, specifically checkpoint inhibitors, which are now at the forefront of cancer treatment. Ongoing trials have been exploring vaccines targeting FRα, mainly in patients with breast and ovarian cancer, in the last few years, but, ultimately, all patients with cancers could potentially benefit.

## 7. Other Uses of FRα

### 7.1. Intraoperative Identification of Tumor

Residual disease after surgery reduces patient survival rates. Therefore, identifying residual or hidden tumors during surgery has become increasingly important. FRα is a protein that is expressed in many different types of tumors. Identifying FRα is essential not only for targeted treatment, but also for improving surgical outcomes. Pafolacianine is an FDA-approved fluorescent drug that binds to FRα. It was first approved by the FDA in 2021 for identifying residual tumors in patients with ovarian cancer during surgery. In a phase III, open-label, 11-center study conducted between March 2018 and April 2020, pafolacianine and near-infrared imaging were used to identify residual cancer tissue in 33% of patients with FRα-positive ovarian cancers undergoing tumor resection. This tissue was not detected by palpation or white light. The sensitivity of detection of ovarian cancer was 83%. Complete R0 resection was achieved in 62.4% of patients. No serious adverse events or deaths were reported during this study [46].

In a phase III ELUCIDATE trial that involved randomization, pafolacianine was used during surgery to treat patients with lung cancer, helping to identify hidden tumors [47]. This study, like the previous study, reported no serious adverse side effects,. This new development is a significant advancement that will enable surgeons to perform less invasive, more precise surgeries. This approach will ensure that the tumor is optimally removed and will lead to improved outcomes.

### 7.2. Nanoparticle Therapies

Nanotechnology has emerged as a promising strategy for the delivery of precise cancer treatments by targeting cancer cells. Such targeting reduces the systemic side effects typically associated with standard chemotherapy. In contrast, chemotherapy non-selectively destroys all cells within the body, resulting in undesirable side effects that may cause patients to discontinue treatment. Modern treatment paradigms have thus shifted their focus from non-selective killing to personalized, targeted approaches such as CAR-T, ADC, BITES, and immunotherapy. This shift has generated public interest in oncology.

FRα, a specific receptor, has a significant advantage when it comes to targeting cancer cells. This receptor is more likely to be expressed in cancer cells than in healthy cells, and it is endocytosed into the cell. The internalization and differential expression of this receptor enable the avoidance of unwanted side effects and the maximization of treatment effects. Developing better therapies through improved delivery systems is therefore one of the fundamental principles of nanoparticle therapies.

Nanoparticles induce apoptosis in cancer cells by targeting them through reactive oxygen species (ROS) [48]. Nanoparticle delivery systems employing FRα have successfully transported chemotherapy drugs such as methotrexate, paclitaxel, and cisplatin to cancer cells. Examples include heparin-folic acid paclitaxel (HFT), liposome-incorporated antisense HER-2, and acetylated generation-5 dendrimers, among others [49,50]. Heparin, paclitaxel, and folic-acid conjugates (HFT-T) have been shown to improve paclitaxel delivery and tumoricidal effects. Nab-paclitaxel conjugated to human albumin and folic acid has been proven effective in targeted delivery to nasopharyngeal and colorectal carcinomas in vitro [50]. Singh et al. generated novel planetary ball milled (PBM) nanoparticles encapsulating resveratrol (RES) and docetaxel and combined with folic acid, which lowered resistance to docetaxel through inhibition of drug efflux [31]. Nanoparticle technology has also explored improved delivery of immunotherapy for melanoma treatment [33]. Chemotherapies that exploit FRα can be made more effective because drug conjugates can target FRα. The examples mentioned above demonstrate this idea. Continued development of drug-delivery systems can bring us closer to improving cancer treatments.

### 7.3. FRα Use as a Prognostic Marker

In 2020, a meta-analysis was conducted on 12 pertinent studies across seven countries, including the United States, Japan, Belgium, Canada, Austria, Netherlands, and Germany [51]. The objective was to determine the correlation between FOLR1 expression and patient outcomes in breast cancer, lung cancer, endometrial cancer, and ovarian cancer. This study found that patients with high FOLR1 expression exhibited poor overall survival (OS) compared to the low-expression group, with a hazard ratio of 0.78 (95% CI: 0.64–0.94, *p* = 0.009) [52]. Furthermore, high FOLR1 expression was correlated with high tumor grade, nodule number, and FIGO stage. This meta-analysis highlighted the prognostic value of FOLR1 across various cancer types, a finding that has been corroborated by more recent studies, including one focusing on rectal cancer [52].

Several studies have reported that patients with triple-negative breast cancer (TNBC) whose cancer cells expressed FRα had improved outcomes, including invasive disease-free survival (IDFS). Norton et al. conducted a study on 384 TNBC patients and found that approximately 71% of their cancers expressed FRα [51]. Adjusting for baseline characteristics, FRα was linked to improved IDFS, but not to improved overall survival (OS). However, other studies have reported different results. Wu et al. investigated circulating tumor cells (CTCs) expressing FRα and found higher levels in patients with breast cancer with metastatic disease and higher tumor stages [53]. Levels of expression decreased significantly post-operatively. However, this study’s sample size was limited, and larger studies are necessary. Similarly, Bax et al. reported that FRα expression patterns were correlated with disease severity in patients with ovarian cancer, with higher expression seen in high-grade serous tumors compared to other types [17]. Cao et al. also found that FRα expression was correlated with poor prognosis in GC and was linked to lymphovascular invasion and extra-nodal disease, indicating it could serve as a useful prognostic tool [27].

Overall, the use of the FRα receptor as a prognostic marker is promising and could revolutionize cancer treatment. Measuring FRα levels across various cancer types may aid in predicting patient outcomes. Patients expressing higher levels may require more aggressive treatment, and oncologists could use this marker to predict tumor behavior and take a more aggressive treatment approach as needed.

### 7.4. Surrogate of Chemotherapy Resistance

Multiple studies have demonstrated that FRα expression is linked to resistance to chemotherapy. For instance, in cases of ovarian cancer, patients whose cancer cells showed higher FRα expression experienced a poor response to chemotherapy, as evidenced by an OR of 8.97 (95% CI: 1.40–57.36, *p* = 0.021) [54]. Notably, patients whose cancers were resistant to chemotherapy were found to have a higher incidence of suboptimal surgery and higher expression of tissue FRα. In this study, FRα was found to suppress the apoptosis that is normally induced by cytotoxic drugs through caspases and apoptosis-related molecules such as Bax and Bcl-2. Additional research has expanded upon this topic, with some studies indicating that certain chemotherapy agents may be more effective in individuals with high levels of FRα expression. For example, in one study, FOLR1 expression was linked to a higher sensitivity to cisplatin treatment [55].

While the mechanism of resistance is not yet fully understood, studies have explored this question. FRα has been linked to resistance to chemotherapy and radiotherapy, with this association potentially explained in part by the activity of MDM2 and p53 [56]. Specifically, MDM2 plays a significant role in chemotherapy resistance, particularly in resistance to cisplatin, as illustrated in Figure 5. MDM2 interacts with p53, which in turn regulates p21, a molecule that has been shown to play a role in chemotherapy resistance [57,58]. In a study of gastric cancer, MDM2 knockout enhanced chemotherapy sensitivity in a way similar to FRα knockouts [59]. In this study, FRα was found to stabilize MDM2 through its binding with a chaperone protein known as PHB2. Notably, MDM2 and FRα knockouts increased the sensitivity of gastric cancer to oxaliplatin.

Further studies are required to evaluate the potential uses of this receptor and its association with chemotherapy resistance. In the future, FRα may be used as a surrogate for chemotherapy resistance, prompting clinicians to consider alternative therapies such as ADC, immunotherapy, or even CAR-T. Alternatively, targeting this receptor may help overcome chemotherapy resistance, particularly in cases of platinum therapy.

### 7.5. Use of FRα in ctDNA

Based on recent medical advances, FRα holds great potential as a target for cancer diagnosis and treatment. While the Ventana FOLR-2.1 diagnostic assay can identify FRα positivity in cancer tissue through immunohistochemistry, it may not always be accessible. Fortunately, ApoStream technology and laser-scanning cytometry using antibodies against FRα have enabled the detection of FRα-positive circulating tumor cells (CTCs) in blood samples across different cancer types, such as NSCLC (*n* = 14), breast cancer (*n* = 20), and ovarian cancer (*n* = 6), which can be compared to samples from healthy subjects (*n* = 20) [60]. This finding suggests that ctDNA/CTCs detection of FRα could be a valuable non-invasive tool for diagnosis, monitoring, and guiding treatment for FRα-expressing cancers.

Moreover, FRα is correlated with EGFR expression in NSCLC, which can be difficult to detect accurately due to variability and insufficient tissue samples. In such instances, FRα ctDNA can serve as a surrogate for EGFR positivity and aid in diagnosis [61]. Furthermore, FRα is more prevalent in adenocarcinomas than in squamous-cell lung cancers, underscoring its potential as a promising tool in ctDNA research for diagnosis, prognosis, treatment, and therapy-response monitoring [62].

## 8. Conclusions

FRα is a receptor that is highly expressed in numerous cancer cells but not in healthy tissues. It is a pivotal determinant of cell growth and tumor proliferation. Not only does FRα play a critical role in novel cancer therapies, but it is also vital for intraoperative identification of occult tumors, vaccine delivery, and CAR-T effectiveness, and it may come to serve as an essential part of precision medicine.

This review aims to expand upon the novel concept of targeted cancer-cell eradication via the utilization of a novel receptor that could potentially pave the way for a better delivery system while avoiding systemic side effects. The future of FRα holds many possibilities due to its associated endocytosis mechanism/internal delivery system. Being able to deliver highly toxic drugs, whether chemotherapy or immunotherapy, to the inside of cancer cells via endocytosis without concerns regarding systemic effects is a pivotal point for the future of cancer treatments.

These drugs could take the form of immunotherapies, antibody-drug conjugates, bispecific antibodies, CAR-T, nanotechnology delivery systems, or other personalized and targeted approaches, steering away from the non-personalized, non-targeted approach of systemic chemotherapy, with its many toxic side effects. As we have transitioned from non-targeted killing to more targeted, personalized medicine, FRα will be pivotal. Not only is its delivery system excellent for causing cancer cells to internalize potentially toxic therapies, but the preferential distribution of this FRα in cancer cells compared to healthy cells potentiates this effect.

Furthermore, with the advent of liquid biopsies, genetics, neo-genomics, and approaches to minimally invasive detection, the detection of this receptor may become more feasible in the coming years. Detection of FRα can be used postoperatively to detect residual tumors, to screen blood samples for cancers, to detect malignancy via immunohistochemistry, to evaluate expression correlating to prognosis/chemotherapy resistance, or to enhance existing therapies. This powerful receptor can form the basis of a new paradigm in the future of cancer care.

## Figures and Tables

**Figure 1 ijms-25-01046-f001:**
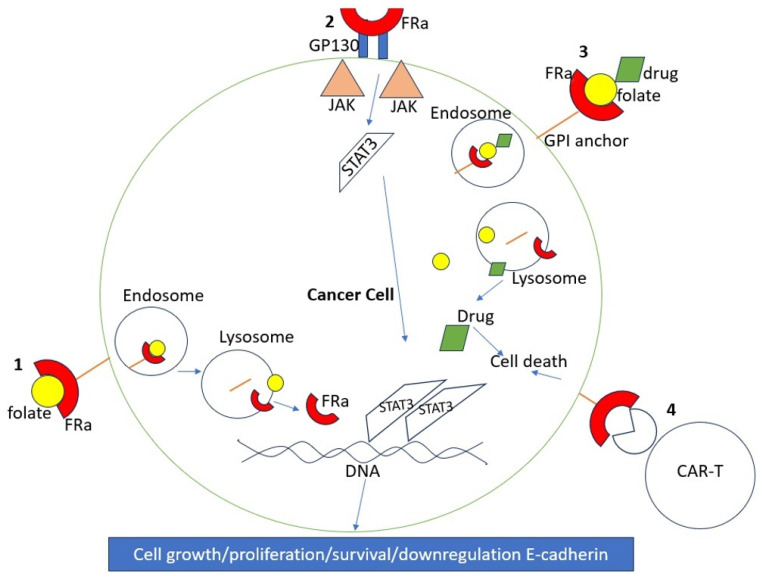
How FRα leads to cell growth. 1. Folate is bound by FRα, which is bound to the membrane via a GPI anchor. Endocytosis occurs, and the acidic pH of the lysosome causes dissociation of the contents. FRα is then free to regulate DNA, leading to cell growth and survival. 2. FRα binds to GP130, leading to activation of the JAK/STAT3 pathway. 3. Example of drug delivery into cancer cells using FRα. 4. Example of CAR-T using FRα.

**Figure 2 ijms-25-01046-f002:**
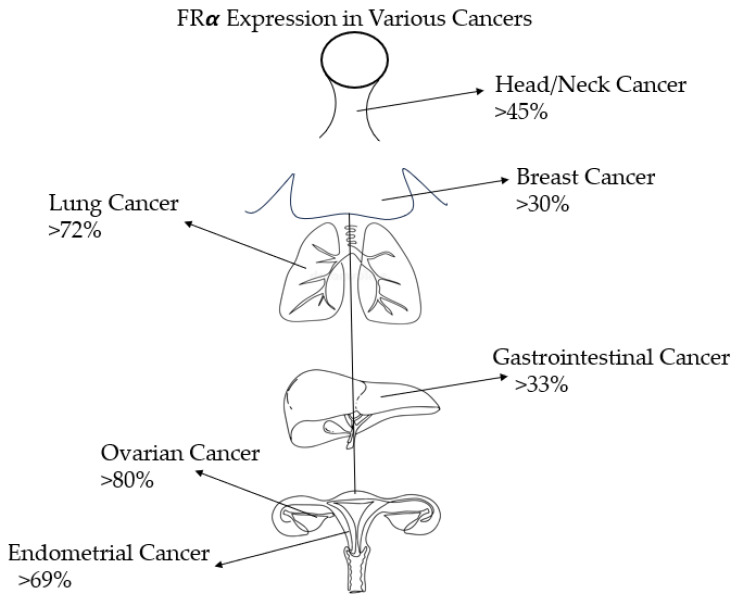
FRα expression levels in various example cancer types.

**Figure 3 ijms-25-01046-f003:**
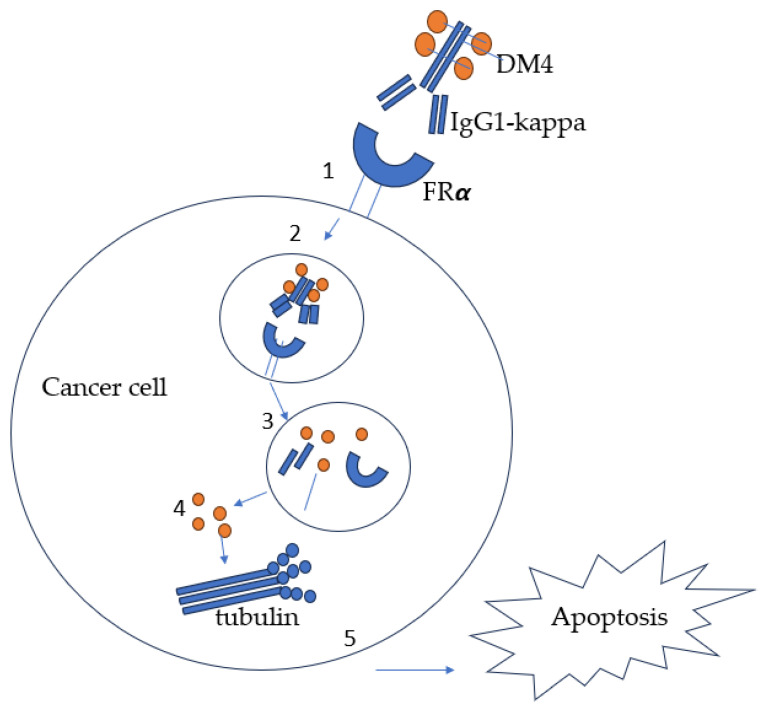
Mechanism of action of MIRV. 1. DM4 linked to IgG1-kappa binds FRα. 2. Endocytosis of antibody-drug conjugate. 3. Lysosomal trafficking and breakdown of payload DM4. 4. DM4 released into the cytoplasm and inhibits tubulin polymerization. 5. Apoptosis of the cancer cell.

**Figure 4 ijms-25-01046-f004:**
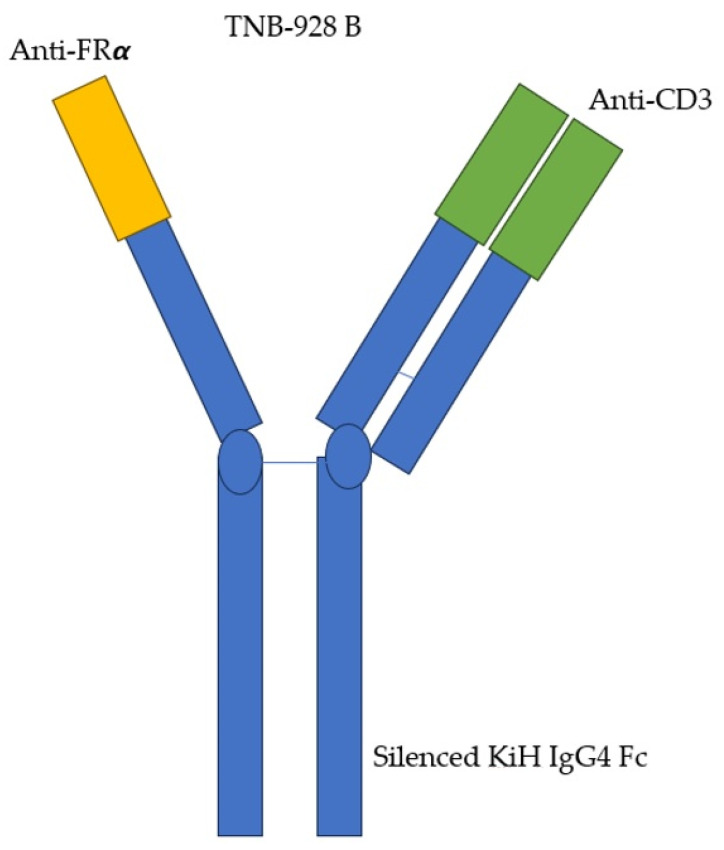
TNB-928B structure: a fully humanized TCE with a bivalent arm for FRα and CD3 used to target FRα-over-expressing cancer cells.

**Figure 5 ijms-25-01046-f005:**
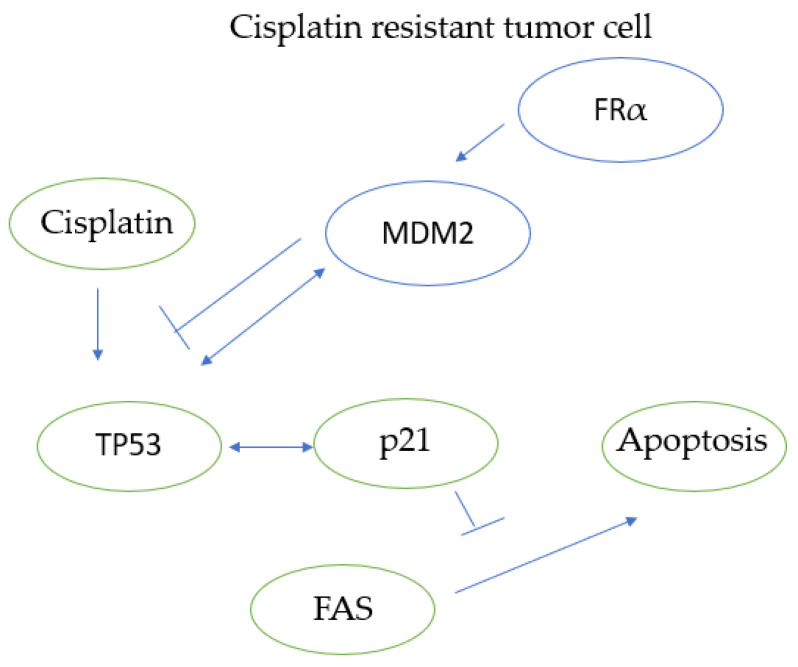
Diagram depicting molecular interactions in cisplatin-resistant tumor cells. FRα stabilizes MDM2, which interacts with TP53. TP53 activation leads to the activation of p21, which in turn inhibits FAS-mediated apoptosis, rendering the tumor cell unable to undergo apoptosis on exposure to cisplatin. Cisplatin, especially in cisplatin-resistant tumor cells, is associated with higher p21 levels via its interaction with TP53, which in turn inhibit FAS-mediated apoptosis.

**Table 1 ijms-25-01046-t001:** FRα expression across various tumor types.

Tumor Type	FR α	Notes
Gastrointestinal Cancers	33–44%	As opposed to only 7% in normal colon tissue or adenomas. More prevalent in younger patients, with increased expression in metastases and cancer cells with non-high-frequency microsatellite instability.
Gynecological Cancers		
Ovarian	80–96%	Mostly present in epithelial ovarian cancers, cancers of the fallopian tubes, and primary peritoneal cancers.
Endometrial	69–81%	Mostly present in serous endometrial cancers
Breast Cancer		
Stage 1–3	~30%	In invasive ductal breast cancers, regardless of their hormonal profile
Triple-Negative Breast Cancer	67–80%	Overexpression is associated with worse outcomes in breast cancer, including decreased disease-free survival
Lung Cancer		
Non-Small Cell Lung Cancer	72–87%	Higher frequency in adenocarcinoma compared to squamous cell carcinoma
Mesothelioma	~72%	Use of the antifolate chemotherapeutic pemetrexed may not necessarily improve the outcome
Head and Neck Cancers	~45%	Presence confers longer survival

**Table 2 ijms-25-01046-t002:** Examples of recent clinical trials utilizing FRα.

Name of Intervention (Study ID)	Years	Phase	Status
Technetium Tc 99m EC20 (NCT01689766)	2003–2007	Phase II	Completed
Folate receptor in stored plasma (NCT00787514)	2008–2011	Observational	Completed
Measurement of the folate receptor in blood (NCT00838747)	2008–2011	Observational	Completed
EC145 as a single-agent therapy and the combination of EC145 plus docetaxel versus docetaxel alone in participants with folate-receptor-positive [FR(++)] second-line NSCLC (NCT01577654)	2011–2015	Phase II	Completed
Mirvetuximab soravtansine (NCT01609556)	2012–2018	Phase I	Completed
Multi-epitope folate receptor alpha peptide vaccine (NCI-2012-00586)	2012–2018	Phase I	Completed
ONX-0801 (a novel α-folate-receptor-mediated thymidylate synthase inhibitor) (NCT02360345)	2013–2021	Phase I	Completed
Multi-epitope folate-receptor-alpha-loaded dendritic cell vaccine (NCT02111941)	2014–2024	Phase I	Active
OTL38 injection (OTL38) (NCT02317705)	2014–2015	Phase II	Completed
Mirvetuximab soravtansine (NCT02606305)	2015–2021	Phase I, II	Completed
Levels of soluble folate receptor and tumor-based folate receptor (NCT02520115)	2015–2018	Phase I	Completed
Folate receptor alpha peptide vaccine (NCT02593227)	2016–2021	Phase II	Completed
Mirvetuximab soravtansine (FORWARD I) (NCT02631876)	2016–2020	Phase III	Completed
Mov18 IgE (NCT02546921)	2016–2021	Phase I	Completed
TPIV200/huFR-1 (A multi-epitope anti-folate-receptor vaccine) (NCT02764333)	2016–2021	Phase II	Completed
[18F]-Azafol as a tracer in positron emission tomography (PET) in folate-receptor-positive cancer (NCT03242993)	2017–2019	Phase I	Completed
Farletuzumab ecteribulin (NCT03386942)	2017–2022	Phase I	Completed
Folate receptor alpha peptide vaccine with GM-CSF as a vaccine adjuvant following oral cyclophosphamide versus GM-CSF (NCT03012100)	2017–2026	Phase II	Active
OTL38 for injection (NCT02872701)	2017–2018	Phase II	Completed
OTL38 (NCT03180307)	2018–2020	Phase III	Completed
Mirvetuximab soravtansine (MIRV) (MIRASOL) (NCT04209855)	2019–2024	Phase III	Active
Mirvetuximab soravtansine (MIRV) (NCT04296890)	2020–2022	Phase III	Completed
OTL38 for injection (NCT04241315)	2020–2021	Phase III	Completed
Mirvetuximab soravtansine (MIRV) (PICCOLO) (NCT05041257)	2021–2024	Phase II	Active
ITIL-306 (NCT05397093)	2022–2039	Phase I	Active

## Data Availability

Not applicable.

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
