# Peer review of "Folate Receptor Alpha—A Novel Approach to Cancer Therapy"

_ijms, 2024, doi:10.3390/ijms25021046_

Round 1
Reviewer 1 Report
Comments and Suggestions for Authors
The manuscript entitles "Folate Receptor Targeting in Cancer Therapy: Current Scientific and Clinical Advancements " submitted by Gonzalez et al., has summarized the various therapeutic and molecular aspects of targeting folate receptors for improved cancer prognosis. The study seems interesting but has been reported to have severe flaws that need to be focused for improving the quality of work. Therefore, the manuscript should not be accepted in its current format and may be considered after major revision.
Major Comments:
1. The introduction section must be improved with a writing sequence focusing on the clarity and need of the current work, as similar works have already been reported.
2. The draft comprises of many sentences that need to be checked and reframed.
3. The writing approach is very generalized. The authors should state connecting statements in the manuscript, which seems to be more justified.
4. The authors have only listed studies in tabular form. Looking the quality and significance of the journal, they should include their thoughts in form of figures. The entire manuscript has been included with varying sections and subsections, but not a single section consists of any figure.
5. Earlier similar review works has also been published with similar earlier discussed studies and approaches (few of them mentioned below):
· Fernández M, Javaid F, Chudasama V. Advances in targeting the folate receptor in the treatment/imaging of cancers. Chem Sci. 2017 Dec 18;9(4):790-810. doi: 10.1039/c7sc04004k. PMID: 29675145; PMCID: PMC5890329.
· Rana A, Bhatnagar S. Advancements in folate receptor targeting for anti-cancer therapy: A small molecule-drug conjugate approach. Bioorg Chem. 2021 Jul;112:104946. doi: 10.1016/j.bioorg.2021.104946. Epub 2021 Apr 27. PMID: 33989916.
Authors must justify how their work is novel or different from the reported studies.
6. Conclusion section need to be more elaborately described highlighting the major significance of the work and how the findings effect for studies in future perspectives.
7. Long sentences have been mentioned but without any specific citation. This is common in various sections throughout the manuscript. Example: no citation found for line no.- 106-112- “This agent is unique in that it achieves rapid renal elimination, important for reducing toxic side effects. After binging to the receptor on the outside of tumor cells, it internalizes into the tumor, goes to the lysosome which releases the exatecan. Not only is this an interesting system of delivery but it is also a study that is not confined to gynecologic malignancies. The study is looking into other malignancies that may express the receptor, such as cholangiocarcinoma, lung, gastric, triple negative breast, colorectal, among others (NCT05001282).”
8. The entire manuscript has English and grammar errors. Language is very difficult to understand.
Comments on the Quality of English Language
The entire manuscript has English and grammar errors. Language is very difficult to understand.
Author Response
1. The introduction section must be improved with a writing sequence focusing on the clarity and need of the current work, as similar works have already been reported.
-Added with an additional paragraph in the introduction (2nd paragraph) to address this
2. The draft comprises many sentences that must be checked and reframed.
Addressed
3. The writing approach is very generalized. The authors should state connecting statements in the manuscript, which seems to be more justified.
Addressed
4. The authors have only listed studies in tabular form. Looking the quality and significance of the journal, they should include their thoughts in form of figures. The entire manuscript has been included with varying sections and subsections, but not a single section consists of any figure.
- Added Figure 1 in the introduction.
5. Earlier similar review works has also been published with similar earlier discussed studies and approaches (few of them mentioned below):
· Fernández M, Javaid F, Chudasama V. Advances in targeting the folate receptor in the treatment/imaging of cancers. Chem Sci. 2017 Dec 18;9(4):790-810. doi: 10.1039/c7sc04004k. PMID: 29675145; PMCID: PMC5890329.
· Rana A, Bhatnagar S. Advancements in folate receptor targeting for anti-cancer therapy: A small molecule-drug conjugate approach. Bioorg Chem. 2021 Jul;112:104946. doi: 10.1016/j.bioorg.2021.104946. Epub 2021 Apr 27. PMID: 33989916.
Authors must justify how their work is novel or different from the reported studies.
-This review provides general updates regarding our knowledge of FRa – it utilizes newer studies, some of them being meta-analyses. A lot of studies are updated from recent years, 2022 and 2023. There’s also information on newer trials, with a table of new/ongoing clinical trials. It provides new topics such as CART, nanoparticle technology, use for intraoperative ID of tumors, use as a prognostic marker, and for chemotherapy resistance. Also focuses in depth on ovarian cancer and briefly expands on current trials across other cancer types. This review discusses Mirvetuximab and expands on the trials leading to its approval. This is more of a review of major approaches in the field of Fra and touches up and interesting uses for the near future, such as leaning away from a nontargeted chemotherapy approach to more personalized medicine.
6. Conclusion section need to be more elaborately described highlighting the major significance of the work and how the findings effect for studies in future perspectives.
-Added more information in the conclusion section addressing this.
7. Long sentences have been mentioned but without any specific citation. This is common in various sections throughout the manuscript. Example: no citation found for line no.- 106-112- “This agent is unique in that it achieves rapid renal elimination, important for reducing toxic side effects. After binging to the receptor on the outside of tumor cells, it internalizes into the tumor, goes to the lysosome which releases the exatecan. Not only is this an interesting system of delivery but it is also a study that is not confined to gynecologic malignancies. The study is looking into other malignancies that may express the receptor, such as cholangiocarcinoma, lung, gastric, triple negative breast, colorectal, among others (NCT05001282).”
-Has been addressed; this is all part of reference number 18
8. The entire manuscript has English and grammar errors. Language is very difficult to understand.
Addressed

Reviewer 2 Report
Comments and Suggestions for Authors
The presented work reviews the potential of the folate receptor as a potential therapeutic target in cancer. The review is potentially interesting, but several concerns should be adressed by the authors.
1: The introduction section, as well as the abstract, do not explain what is the overall purpose of the review. The authors should add it for both sections.
2: The authors should expand and talk deeper about the structure and function of FRα. Maybe add a figure to easier explain the pathways that the receptor can activate.
3: The authors deeply analyze the role of FRα in ovarian cancer, dedicating a whole section to it, while summarizing its importance in other cancer types in another paragraph. Altough this is correct and acceptable, the authors should explain why they focus in particular on ovarian cancer. Moreover, a figure or a schematic diagram highlighting the function of FRα in other cancers would improve the readability of the paper.
4: The authors should explain, or at least speculate, the molecular mechanism behind the resistance pattern of FRα when used for CAR therapy.
5: Similarly, the importance of FRα for resistance to chemotherapy should be deeper analyzed, and the molecular mechanisms behind this should be explained. Is there only one reference about FRα as a mediator of resistance?
6: The conclusion should include additional discussion about the direction of the future research about FRα for cancer therapy.
7: My biggest concern about the manuscript regards its structure and writing. Several parts should be rephrased, or made more clear. Below there is a short list, but all the manuscript should be deeply proof-read and improved on the writing.
Lanes 27, 32, 37, 52-54, 63-64, 76-79, 123-124, 129-130, 141-142, 154, 157-160, 180-181,251-254: i found the sentences in those lanes unclear, and they should be improved and re-phrased.
Lane 48-49, 66-69, 111, 269-282: In all those parts an appropriate reference should be added at the end of the sentence.
Lane 84: Reference should be moved to lane 89.
Comments on the Quality of English LanguageThe English quality of the paper is acceptable, except where the sentences should be re-phrased.
Author Response
1: The introduction section, as well as the abstract, do not explain what is the overall purpose of the review. The authors should add it for both sections.
-This has been addressed in the introduction section
2: The authors should expand and talk deeper about the structure and function of FRα. Maybe add a figure to easier explain the pathways that the receptor can activate.
-This has been addressed by adding a figure
3: The authors deeply analyze the role of FRα in ovarian cancer, dedicating a whole section to it, while summarizing its importance in other cancer types in another paragraph. Altough this is correct and acceptable, the authors should explain why they focus in particular on ovarian cancer. Moreover, a figure or a schematic diagram highlighting the function of FRα in other cancers would improve the readability of the paper.
-Addressed why the focus is on ovarian cancer in the first paragraph of the ovarian cancer section
-Function would be difficult to explain in an image – table 1 has the different expressions across cancer types and introduction has information about general function of the receptor
4: The authors should explain, or at least speculate, the molecular mechanism behind the resistance pattern of FRα when used for CAR therapy.
-We addressed some reasons why CAR-T therapy may become ineffective. We focused more on targeting the FRa as a new target and more on how it can improve immune responses. FRa has been associated with chemotherapy resistance so we discuss that more.
5: Similarly, the importance of FRα for resistance to chemotherapy should be deeper analyzed, and the molecular mechanisms behind this should be explained. Is there only one reference about FRα as a mediator of resistance?
-This has been addressed and expanded further. We discuss a possible resistance mechanism. More studies are needed to address this question.
6: The conclusion should include additional discussion about the direction of the future research about FRα for cancer therapy.
-This has been addressed by adding more throughout the paper about future perspectives.
7: My biggest concern about the manuscript regards its structure and writing. Several parts should be rephrased, or made more clear. Below there is a short list, but all the manuscript should be deeply proof-read and improved on the writing.
Lanes 27, 32, 37, 52-54, 63-64, 76-79, 123-124, 129-130, 141-142, 154, 157-160, 180-181,251-254: i found the sentences in those lanes unclear, and they should be improved and re-phrased.
Lane 48-49, 66-69, 111, 269-282: In all those parts an appropriate reference should be added at the end of the sentence.
Lane 84: Reference should be moved to lane 89.
-We have re-written the entire paper to make it easier to understand/read, sentences have been rephrased.

Reviewer 3 Report
Comments and Suggestions for Authors
The review manuscript addresses the topic of Folate Receptor Targeting in Cancer Therapy, specifically focusing on Current Scientific and Clinical Advancements. I do not recommend its publication for the following reasons:
1. There is a lack of coverage regarding the history of folate receptor function in various cancers, including the role of other folate receptors and their distinctions.
2. The references used in the manuscript are only partially explained or discussed within the text, resembling more of a report on scientists' findings rather than providing a comprehensive understanding of the advancements and future directions in this field.
3. The mechanism of how folate receptor endocytosis enables the nanoparticle/drug delivery containing the folic acid substrate to cancer cells instead of healthy cells is not adequately explored.
4. The reviewed manuscript includes very limited scientific research, which is deemed unacceptable. It predominantly focuses on well-known therapies, providing minimal discussion on less-covered topics that could be more beneficial for readers.
5. The scope of the review is unclear in my opinion and leaving readers without a clear understanding of the manuscript's overarching purpose.
Due to these deficiencies, I believe the manuscript is not suitable for publication.
Comments on the Quality of English LanguageModerate English editing is needed.
Author Response
1. There is a lack of coverage regarding the history of folate receptor function in various cancers, including the role of other folate receptors and their distinctions.
-The focus of our paper is to address FRa subtype specifically. We briefly did mention the history of targeted agents for FRa in the section about Mirvetuximab soravtansine (MIRV) in the first sentences. The function of FRa is explained in the introductory section.
2. The references used in the manuscript are only partially explained or discussed within the text, resembling more of a report on scientists' findings rather than providing a comprehensive understanding of the advancements and future directions in this field.
-Addressed. Elaborated on future advancements/directions.
3.The mechanism of how folate receptor endocytosis enables the nanoparticle/drug delivery containing the folic acid substrate to cancer cells instead of healthy cells is not adequately explored.
-Addressed. A figure was provided to address this.
4. The reviewed manuscript includes very limited scientific research, which is deemed unacceptable. It predominantly focuses on well-known therapies, providing minimal discussion on less-covered topics that could be more beneficial for readers.
- We expanded on recent literature and addressed the major topics pertaining to FRa
5. The scope of the review is unclear in my opinion and leaving readers without a clear understanding of the manuscript's overarching purpose.
-Made the purpose of our paper clearer in the introductory section, and conclusion. Also tied the topics to future perspectives.

Round 2
Reviewer 1 Report
Comments and Suggestions for Authors
The revised draft of the manuscript entitled "Folate Receptor Targeting in Cancer Therapy: Current Scientific and Clinical Advancements" seems good to me. Still, I have few suggestions that the authors can consider for improving the quality of work. I have mentioned few comments that need to be answered before acceptance of the manuscript.
1. The initial section of the manuscript should be included with a paragraph highlighting the epidemiological, pathogenesis and treatment approaches for cancer worldwide.
2. The introduction section should also show why you have focused specifically on folate receptors for targeting cancer. Is there any specific significance and how current therapies are working on it?
3. The authors should insert few illustrations/figures in section 4, 5, 6 and 7 showing the research highlights of the work cited in the manuscript.
4. The authors should include some important statements for the future perspectives of targeting folate receptors in cancer therapy, in the "conclusion" section.
5. Look for the overall grammatical, scientific statements and English language quality in the manuscript before final submission.
Comments on the Quality of English LanguageModerate editing of English language required
Author Response
- The initial section of the manuscript should be included with a paragraph highlighting the epidemiological, pathogenesis and treatment approaches for cancer worldwide. [corrected]
2. The introduction section should also show why you have focused specifically on folate receptors for targeting cancer. Is there any specific significance and how current therapies are working on it? [corrected]
3. The authors should insert few illustrations/figures in section 4, 5, 6 and 7 showing the research highlights of the work cited in the manuscript. [corrected]
4. The authors should include some important statements for the future perspectives of targeting folate receptors in cancer therapy, in the "conclusion" section. [corrected]
5. Look for the overall grammatical, scientific statements and English language quality in the manuscript before final submission. [corrected]
Reviewer 2 Report
Comments and Suggestions for Authors
Most of my comments were adressed by the authors. Only few minor concern remains.
1) The introduction section has been correctly modified. However, the abstract should be corrected as well.
2) I still believe that Table 1 could be converted into a figure or a diagram, in order to facilitate the reader. If possible, this should be done.
3) The position and explanation of Table 2 remains a bit confusing. It should be better contextualized in the text (maybe mentioning it the first time clinical trials are introduced?)
Comments on the Quality of English Language
Altough greatly improved, the paper still needs minor proofreading for the English quality.
Author Response
1) The introduction section has been correctly modified. However, the abstract should be corrected as well. [corrected]
2) I still believe that Table 1 could be converted into a figure or a diagram, in order to facilitate the reader. If possible, this should be done. [corrected]
3) The position and explanation of Table 2 remains a bit confusing. It should be better contextualized in the text (maybe mentioning it the first time clinical trials are introduced?) [corrected]